# Characterization of congenital hyperinsulinism in Argentina: Clinical features, genetic findings, and treatment outcomes

Gabriela Pacheco[1], Maria G. Bastida[2], Juan Cáceres[3], Guillermo Alonso[4], Mariana Aziz[5], Martha Suarez[6], Adriana Flores[7], Victoria Femenia[8], María V. Forclaz[9], Jayne A.L. Houghton[10,11], Jasmin J. Bennett [10], Sabrina Martin[8], Sarah E. Flanagan [10]*, Ana Tangari-Saredo[12]

1 Pediatric Nutrology, Hospital Público Materno Infantil de Salta, Argentina, 2 Hospital Provincial Neuquén, Dr E Castro Rendón, Neuquén, Argentina, 3 Endocrinology, Hospital Plottier, Neuquén, Argentina, 4 Pediatric Endocrinology Section, Hospital Italiano de Buenos Aires, Buenos Aires, Argentina, 5 Department of Endocrinology, Hospital de Pediatría J.P. Garrahan, Buenos Aires, Argentina, 6 Department of Medicine, Endocrinology Unit, Hospital T. Alvarez, Buenos Aires, Argentina, 7 Nutrition and Diabetes Service, Fundación Hospitalaria, Buenos Aires, Argentina, 8 Endocrinology, Hospital H. Notti, Mendoza, Argentina, 9 Pediatric Endocrine Service, Hospital Posadas, Buenos Aires, Argentina, 10 Department of Clinical and Biomedical Science, University of Exeter, Exeter, United Kingdom, 11 The Genomics Laboratory, Royal Devon University Hospital NHS Foundation Trust, Exeter, United Kingdom, 12 Division of Endocrinology, Sanatorio Güemes, Buenos Aires, Argentina

* S.Flanagan@exeter.ac.uk

## Abstract

### Introduction

Congenital hyperinsulinism (CHI) is a heterogeneous disorder of insulin dysregulation, leading to hypoglycemia. This study describes the clinical characteristics, genetics, and management of CHI in Argentina.

### Methods

We retrospectively reviewed 70 probands diagnosed with CHI (2008–2021) at multiple centres across Argentina. Clinical, biochemical, imaging, and treatment data were analyzed. Genetic testing was performed in 49 probands using Sanger and targeted next-generation sequencing of CHI-related genes.

### Results

Transient CHI was identified in 23/70 (33%) probands, with a median duration of 2 months. Risk factors for perinatal stress-induced hyperinsulinism (PSHI) were present in 85% of transient cases. Persistent CHI was diagnosed in 44/70 (63%) individuals, of whom 31 responded to diazoxide. Late-onset CHI (diagnosed >3 years) was identified in 3 children.

**Data availability statement:** All non-clinical data analyzed during this study are included in this published article. Clinical and genotype data can be used to identify individuals and are therefore available only through collaboration to experienced teams working on approved studies examining the mechanisms, cause, diagnosis and treatment of diabetes and other beta cell disorders. Requests for collaboration will be considered by a steering committee following an application to the Genetic Beta Cell Research Bank https://www.diabetesgenes.org/current-research/genetic-beta-cell-research-bank/). Contact by email should be directed to rduh.betacellgenomics@nhs.net.

**Funding:** This research was funded in whole, or in part, by Wellcome [223187/Z/21/Z]. For the purpose of open access, the author has applied a CC BY public copyright license to any Author accepted Manuscript version arising from this submission.

**Competing interests:** The authors have declared that no competing interests exist.

A pathogenic variant was detected in 19/49 (39%) probands, all had persistent CHI. *ABCC8* variants were most common accounting for 68% (13/19) of diagnoses. Imaging in 17 cases revealed focal disease in 8, diffuse disease in 8, and atypical disease in 1 individual. Seven individuals with focal disease underwent lesionectomy, which was curative in 5 (71%). Three children with diffuse disease required near-total pancreatectomy, with one developing postoperative diabetes.

## Conclusions

This study provides the largest CHI cohort reported from South America and highlights the clinical and genetic heterogeneity of the condition. Transient CHI was often associated with PSHI risk factors, while persistent CHI was predominantly linked to K-ATP channel variants. The findings underscore the importance of genetics and imaging for CHI management and emphasize the need for increased access to molecular diagnostics.

## Introduction

Congenital hyperinsulinism (CHI) causes severe, persistent hypoglycemia. It is diagnosed when there is increased insulin action and/or inadequate suppression of plasma insulin during spontaneous or fasting-induced hypoglycemia [1]. In some cases, CHI may be secondary to perinatal factors such as intrauterine growth restriction (IUGR), maternal diabetes mellitus, or perinatal asphyxia [2]. The hypoglycaemia associated with perinatal stress-induced hyperinsulinism (PSHI) typically resolves within the first weeks of life but can persist for several months in severe cases [3].

CHI that persists beyond six months affects around 1 in 28,000 live births and is often genetic [4,5]. More than 30 genetic forms of CHI have been reported, causing either isolated or syndromic disease [6,7]. Loss-of-function variants in the *ABCC8* and *KCNJ11* genes, which encode the pancreatic ATP-sensitive potassium (K-ATP) channel, are the most common cause of isolated CHI, accounting for ~50% of cases. [8–10] Other common genetic causes of persistent CHI include *GLUD1* variants [11], and variants affecting the hexokinase genes, *GCK* and *HK1* [12,13].

The genetic cause of CHI determines whether pancreatic histology is diffuse, focal, or atypical [1]. A focal lesion arises when a paternally inherited *ABCC8* or *KCNJ11* pathogenic variant is unmasked by paternal uniparental isodisomy in pancreatic tissue [14–16]. In focal CHI, imaging is crucial for localising the lesion before lesionectomy, which is curative in up to 97% of cases [17,18]. Diffuse disease results from biallelic or dominant variants in the CHI genes. Those with diffuse CHI who are unresponsive to maximal medical therapy, including diazoxide or somatostatin analogues, may require near-total pancreatectomy which carries a high risk of permanent diabetes and exocrine insufficiency [17,19,20]. Atypical histological disease has been reported in a few individuals with somatic mosaic variants [21–23].

There is limited data on the genetics and clinical management of CHI in South America. This study aims to describe our experience in managing children with this heterogeneous condition in Argentina.

## Materials and methods

### Subjects

Clinical and genetic data were retrospectively collected from clinicians managing children diagnosed with CHI at centres across Argentina between 01/01/2008 and 31/12/2021. The data were accessed for research purposes between 01/01/2023 and 31/12/2024 with the analysis of the data conducted jointly between the consulting centres. CHI was diagnosed based on detectable plasma insulin during hypoglycaemia (<60 mg/dl). When available, additional biochemical markers, including beta-hydroxybutyrate (<1.8 mmol/L) and free fatty acids (<1.7 mmol/L), along with an inappropriate increase in blood glucose (> 30 mg/dL) following parenteral glucagon administration, were used to support the clinical diagnosis of CHI. Individuals with evidence of syndromic disease at the time of CHI diagnosis were excluded.

The study complied with the Declaration of Helsinki, with informed written consent obtained from the parents of all patients. The study was approved by the Wales Research Ethics Committee 5 (22/WA/0268), with participants recruited to the Genetic Beta Cell Research Bank (IRAS: 316050).

Transient CHI was defined as full remission by six months of age, confirmed by cessation of all treatment. CHI persisting beyond six months was classified as persistent and further subdivided based on diazoxide response. Diazoxide-responsive individuals maintained normal plasma glucose levels for at least 4.5 to 6 hours during a fasting test, while those on 15 mg/kg/day of diazoxide who failed to sustain glucose above 70 mg/dL for 6-hours were classified as diazoxide-unresponsive. Late-onset was defined as CHI diagnosed after 3 years.

### Molecular genetics

Genetic testing was performed on 49 probands. Of these, 43 were tested at the Exeter Genomics Laboratory and six received testing through a commercial laboratory. Of the 49 children who underwent testing, 7 had transient CHI, 39 had persistent CHI and 3 had late-onset CHI.

In Exeter, Sanger sequencing was the first-line test when clinical features suggested a specific genetic subtype (e.g., *GLUD1* in individuals with hyperammonaemia) [11]. For the remaining individuals, and those without variants identified by Sanger sequencing, targeted next-generation sequencing (tNGS) of 15 CHI genes (*KCNJ11, ABCC8, GLUD1, GCK, HADH, HK1, HNF1A/4A, INSR, KDM6A, KMT2D, SLC16A1, CACNA1D, PMM2, TRMT10A*) was performed as previously described [24]. For *HK1* and *SLC16A1,* screening was limited to the previously described non-coding regulatory regions where disease-causing variants have been identified [13,25,26]. Next generation sequencing analysis included on- and off-target copy number variant analysis using the in-house software SavvyCNV (methods previously described [27]). Parents and clinically affected family members were tested by Sanger sequencing and variants were classified according to established guidelines [28,29].

### Imaging

[18]F-DOPA PET-CT imaging was performed on a Philips Gemini True Flight Technology 64-detector-row machine. The images were reconstructed in axial, sagittal and coronal projections and interpreted both with and without attenuation corrected. Early abdominal images were acquired at 30 minutes, while late whole-body images were acquired 60 minutes post-injection.

### Statistical analysis

ANOVA was used to compare quantitative variables, and the Kruskal-Wallis test was applied to normally distributed data. *P*-values <0.05 were considered significant.

## Results

A total of 70 probands were identified. Consanguinity was not reported in any of the families. In 23 of 70 (33%) children, CHI remitted before six months, with an average disease duration of 2 months [IQR: 0.5–6 months]. Nineteen of the 23 children (83%) with transient CHI had one or more risk factors for PSHI, including IUGR (n = 8), gestational hypertension and pre-eclampsia (n = 5), gestational diabetes (n = 4), prematurity (gestational age < 37 weeks) (n = 6), fetal bradycardia (n = 1), or erythroblastosis (n = 1). Sixteen children with transient CHI were treated with diazoxide (mean dose 8.4 mg/kg/day) without adverse effects, while 7 were managed with IV glucose and feeding strategies.

CHI was diagnosed in early infancy and persisted beyond six months in 44 of 70 (63%) children. Of these, 31 (70%) responded well to diazoxide. Treatment was discontinued in eight children between 7 months and 8 years following CHI resolution. In the remaining 23, diazoxide dose was reduced from a mean of 10.8 to 7.2 mg/kg/day after a median of 34 months. Adverse effects of diazoxide were observed in four children (receiving 10–12 mg/kg/day), including fluid overload (two without prophylactic diuretics) and, in two cases, neutropenia or thrombocytopenia. In one child, fluid overload resolved after a reduction in the diazoxide dose and the introduction of octreotide therapy. In the remaining three, adjusting the diuretic dose resolved the fluid excess. When patients did not respond to diazoxide, treatment with octreotide was initiated (median dose: 21 ug/kg/day, range: 9–48 ug/kg/day) and/or frequent feeding. One of these children developed gallstones at 13 months.

CHI was diagnosed in three individuals after the age of three years. An MRI ruled out insulinoma in all cases. One child, diagnosed with CHI at eight years old, had a history of a seizure at the age of one. A second patient, clinically diagnosed at 14 years, reported episodes of weakness and pallor that resolved with sugar intake starting at the age of four. The third patient, clinically diagnosed with CHI at 8 years old, had no history suggestive of undiagnosed hypoglycemia.

## Genetics

A disease-causing variant was identified in 39% of those tested (n = 19/49). All 19 children had persistent CHI and one had late-onset CHI (case 14, Table 1). Variants were found in the *ABCC8* (n = 13), *GLUD1* (n = 2), *INSR* (n = 1) and *HK1* (n = 1) genes. In two patients a large deletion on the X chromosome or chr20p11.2 region was identified (Table 1). Clinical and genetic data of the individual with the chr20p11.2 deletion have been reported previously [30]. Briefly, this patient was diagnosed with CHI at 52 weeks of age. Hyperinsulinism was ongoing at the age of 3 years, and they were noted to have facial dysmorphism and developmental delay. The previously unreported female patient with an X chromosome deletion was diagnosed with CHI at 39 weeks of age. No syndromic features had been noted at the time of referral for genetic testing.

In seven children, the disease-causing variant had arisen *de novo* or co-segregated with CHI in the family, consistent with a dominant variant (*ABCC8* n = 4, *GLUD1* n = 1, *HK1* n = 1, ChrX deletion n = 1). Compound heterozygous *ABCC8* variants were identified in three probands. In four children, a paternally inherited recessive *ABCC8* variant was found, and in two children, the variant was inherited from an unaffected parent (*INSR* and chr20del). For three children, the inheritance could not be established as parental samples were unavailable (*GLUD1* n = 1, *ABCC8* n = 2). Additionally, in 5 children (10%, n = 5/49), a variant of uncertain significance (VUS) was identified (*ABCC8* n = 3, *KCNJ11* n = 2) (Table 1).

Risk factors for PSHI were identified in 10 of the 30 individuals without a genetic diagnosis. This included 6 individuals with transient CHI and 4 individuals with persistent CHI

## Pancreatic imaging and surgical treatment

Imaging of the pancreas in 17 individuals with persistent CHI demonstrated diffuse tracer uptake in 8, focal uptake in 8 and atypical uptake in 1 (Table 1). Four of the eight children with diffuse uptake had undergone genetic testing prior to imaging. In three of these cases, the genetic results were consistent with diffuse disease, while one child had a paternally

**Table 1. Genetic and clinical data for 24 probands with congenital hyperinsulinism and variants in a known hyperinsulinism gene.**

| Patient ID | Gene | Variant | Inheritance | 18F-DOPA/PET-CT performed? | Diazoxide Response | Surgery performed | Current Medical treatment |
|---|---|---|---|---|---|---|---|
| **Disease-causing Variants** | | | | | | | |
| 1 | ABCC8 | p.(Glu9Ter)/p.(Arg168His) c.25G>T/c.503G>A | Bi-allelic | Not performed | R | No | Diazoxide |
| 2 | ABCC8 | p.(Leu81Pro)/p.(Asn1481fs) c.242T>C[a]/c.4440del | Bi-allelic | Diffuse | NR | No | Octreotide |
| 3 | ABCC8 | p.(Tyr179Ter) c.536_539del | Paternal | Focal | NR | Yes | No medical treatment post-surgery |
| 4 | ABCC8 | p.(Gly505Arg) c.1513G>C | De novo | Not performed | R | No | Diazoxide up to 1 year then lost to follow-up |
| 5 | ABCC8 | p.(Leu508Pro) c.1523T>C | Maternal | Not performed | R | No | Diazoxide |
| 6 | ABCC8 | p.(Arg598Ter) c.1792C>T | Paternal | Focal | NR | Yes | No medical treatment post-surgery |
| 7 | ABCC8 | p.(Arg934Ter)/p.(Arg1215Gln) c.2800C>T/c.3644G>A | Bi-allelic | Not performed | NR | No | Octreotide up to 13 months |
| 8 | ABCC8 | p.(Leu1171fs) c.3512del | Paternal | Diffuse | NR | Yes | Insulin for diabetes post-surgery |
| 9 | ABCC8 | p.(Asp1193fs) c.3577del | Paternal | Focal in head of pancreas | NR | No | Octreotide+diazoxide up two years |
| 10[b] | ABCC8 | p.? c.3653+1G>A | Unknown | Focal | NR | Yes | No medical treatment post-surgery |
| 11[b] | ABCC8 | p.(Gly1379Asp) c.4136G>A | De novo | Diffuse | R | No | Diazoxide up to 8 years |
| 12[b] | ABCC8 | p.(Gly1479Arg) c.4435G>A | Unknown | Diffuse | R | No | Octreotide/lanreotide due to adverse effects of diazoxide |
| 13 | ABCC8 | p.(Ser1483Asn) c.4448G>A | De novo | Not performed | NR | No | Octreotide |
| 14 | GLUD1 | p.(Ile497Met) c.1491A>G | Unknown | Not performed | R | No | No treatment |
| 15 | GLUD1 | p.(Ser498Leu) c.1493C>T | De novo | Not performed | R | No | Diazoxide |
| 16 | HK1 | Chr10:71,108,645T>C | Maternal | Not performed | R | No | Diazoxide |
| 17 | INSR | p.(Arg1191Gln) c.3572G>A | Maternal | Not performed | R | No | Diazoxide |
| 18[c] | Chr20p11.2del | Chr20:19,507,014–22,525,896del | Maternal | Not performed | R | No | Diazoxide |
| 19 | Turner syndrome | ChrX monosomy | De novo | Diffuse | NR | Yes | Diazoxide post-surgery |
| **Variants of uncertain clinical significance** | | | | | | | |
| 20[b] | ABCC8 | p.(Gly716Cys) c.2146G>T | Unknown | Diffuse | R | No | Diazoxide |
| 21 | ABCC8 | p.(Val802Asp) c.2405T>A | Paternal | Diffuse | NR | Yes | No medical treatment post-surgery |
| 22[b] | ABCC8 | p.(Glu1208Lys) c.3622G>A | Unknown | Not performed | R | No | Diazoxide |
| 23 | KCNJ11 | p.(Arg206Cys) c.616C>T | Maternal | Not performed | R | No | Diazoxide up to 5 years |
| 24 | KCNJ11 | p.(Tyr268His) c.802T>C | Paternal | Not performed | R | No | Diazoxide up to 7 months |

All 19 individuals with a disease-causing variant had persistent CHI. R=responsive to diazoxide, NR=not responsive to diazoxide.

[a]This individual has a paternally inherited pathogenic ABCC8 variant and a maternally inherited variant currently classified as of uncertain clinical significance (p.(Leu81Pro), c.242T>C).

[b]Genetic testing performed at a commercial laboratory.

[c]Patient and genetic results previously reported in reference [30].

inherited recessive variant predicting a focal lesion. One additional child with diffuse disease underwent genetic testing following imaging and surgery which confirmed Turner syndrome. The child with atypical uptake had undergone genetic testing, but no disease-causing variant was found.

Three of the children with diffuse disease (n = 1 dominant *ABCC8*, n = 1 *ABCC8* VUS and n = 1 Turner syndrome) and the child with atypical tracer uptake underwent pancreatectomy due to poor response to diazoxide. A second surgery was required in two patients because of persistent hypoglycemia. One child developed diabetes immediately post-surgery, while the other developed chyloperitoneum and exocrine insufficiency.

In the focal group, 4 children had undergone genetic testing prior to imaging which had identified a heterozygous *ABCC8* variant (3 paternal, 1 unknown) (Table 1). Lesionectomy was performed in 7 of the 8 children. In the remaining case, imaging revealed the lesion was in the head of pancreas, this patient was managed with octreotide and frequent feeding. Two of the focal cases developed hyperinsulinism post-surgery, which responded to octreotide. Diabetes or exocrine insufficiency was not reported in any of these cases.

### Clinical characteristics according to persistence of CHI and responsiveness to diazoxide

Patients with persistent diazoxide-unresponsive CHI had higher birthweights than those with persistent diazoxide-responsive or transient CHI (median Z-score 2.2 SDS vs 0.4 SDS vs −0.6 SDS respectively, ANOVA p < 0.001). Insulin levels were also higher at presentation of disease in those who were unresponsive to diazoxide (median insulin level: 22.7 mU/l vs 12.2 mU/l (diazoxide-responsive) vs 11.0 mU/l (transient CHI) ANOVA p = 0.018). Among patients with persistent CHI, those responsive to diazoxide were older at presentation than those with diazoxide-unresponsive (median 30 days vs 1 day, Kruskal Wallis p = 0.001), or transient CHI (median 30 days vs 1 day, Kruskal Wallis p < 0.001).

### Discussion

We describe 70 probands diagnosed with CHI over a 13-year period in Argentina, making this the largest study to date reporting on the diagnosis and management of CHI in South America [31,32].

Transient CHI was present in 33% (23/70) of individuals. This was characterized by an early presentation and a good response to diazoxide without side effects. Unlike other studies, there was no male predominance in the transient CHI cohort (52% male) [3,9,33]. At least one PSHI risk factor was detected in 83% of the transient CHI cases which aligns with previous reports [9,33,34]. Four individuals with persistent CHI also had PSHI risk factors, including one infant with CHI remission at 7 months, who was born small for gestational age following a pregnancy affected by pre-eclampsia.

Genetic testing was performed on 7 individuals with transient CHI, all of whom had risk factors for PSHI, but no disease-causing variants were identified. This is consistent with other studies, which have shown that a genetic diagnosis is unlikely in individuals with transient CHI [9]. Interestingly, the child whose CHI remitted at 7 months of age had a paternally inherited *KCNJ11* VUS, raising the possibility that this may be acting as a phenotypic modifier, potentially causing a prolonged form of PSHI.

Genetic analysis is limited in Argentina, so for 43 families, genetic testing was conducted in the UK through the Open Hyperinsulinism Genes Project, which is funded by the global charitable organization Congenital Hyperinsulinism International [35]. A genetic diagnosis was obtained for 19 of the 49 (39%) patients who underwent testing. This yield is slightly lower than that reported in other large series, likely reflecting the composition of the cohort [8,10,36–39]. Specifically, 7 patients (14%) had transient CHI, 10 individuals had identifiable risk factors for PSHI, including 6 with transient CHI and 4 with persistent HI, and 3 patients had later-onset CHI which is typically associated with a lower diagnostic yield compared to infancy-onset disease [40].

All individuals with a genetic diagnosis had persistent CHI, with *ABCC8* variants accounting for 68% of diagnoses (n = 13/19). No homozygous variants were identified, consistent with the absence of consanguinity in the cohort. In two individuals finding a large contiguous gene deletion confirmed a syndromic form of CHI [30,41]. Neither of these patients

exhibited syndromic features at CHI diagnosis, although additional symptoms were diagnosed later in childhood which were consistent with the syndrome.

Sixteen children with transient CHI and 31 with persistent CHI were treated with diazoxide. This included an individual with a compound heterozygous *ABCC8* variant (p.(Glu9Ter)/p.(Arg168His)), an uncommon feature of biallelic variants in this gene [42–44]. Four individuals experienced adverse effects to diazoxide, which were managed by reducing the dose and/or introducing adjunct therapy and one child treated with octreotide developed gallstones, a recognized side effect of this drug [45,46].

Pancreatic imaging was performed on 17 individuals, 9 of these had received a genetic diagnosis prior to imaging which had correctly predicted the histology in 8 cases (Table 1). In the remaining case with diffuse uptake of the tracer, the presence of a paternal heterozygous variant had predicted focal disease. For this individual it is possible that there is a second, undetected variant, on the maternal allele or the patient had a giant focal lesion that appeared like diffuse disease on imaging [47].

In this study, some patients who had not undergone genetic testing underwent scanning in an effort to avoid prolonged medical treatment. In Argentina and most Latin American countries, imaging is generally more accessible than genetic testing, with 18F-DOPA PET-CT introduced in Argentina in 2014. This greater accessibility is largely due to the wider acceptance of imaging for clinical decision-making and its routine coverage by most private health insurance plans. In contrast, genetic testing is usually a non-listed procedure, requiring individual approval from health insurance providers, requests that are often denied. Access is further limited for patients receiving care outside the capital city, where genetic testing resources are even more limited. With the recent availability of charity-funded genetic testing, prioritizing *ABCC8* and *KCNJ11* screening for future diazoxide-unresponsive cases born in Argentina should be pursued in accordance with current guidelines [1,48].

Seven individuals with a focal lesion underwent lesionectomy. While all were diazoxide-unresponsive at surgery, one had initially responded to diazoxide (10 mg/kg/day) after a 10-hour fast, but later lost response. Surgery was curative in 71% (5/7) of cases, which is slightly lower than other reports [17]. The child with a focal lesion in the pancreatic head and the two others who remained hypoglycemic post-surgery were treated with octreotide, with good response, similar to previous studies [49,50]. In two cases, the octreotide dose was later reduced, indicating decreased disease severity [51].

This retrospective study has some limitations. First, clinical data were collected from the case notes of patients managed at different tertiary centers across Argentina, this highlighted variability in the approach to managing the condition and incomplete data in some cases. Additionally, not all children with persistent CHI underwent genetic testing, and family member samples were sometimes unavailable to assess inheritance. Ensuring access to genetic testing for these families will be crucial to help inform on recurrence risk.

In conclusion, our study highlights the clinical and genetic heterogeneity of CHI in Argentina and underscores some of the many challenges clinicians face in managing this complex condition.

## Acknowledgments

SEF has a Wellcome Trust Senior Research Fellowship (Grant Number 223187/Z/21/Z). We are grateful to Congenital Hyperinsulinism International (a501(c)3 organization) who funded the genetic testing conducted in Exeter for individuals with hyperinsulinism in Argentina through the Open Hyperinsulinism Genes Project.

## Author contributions

**Conceptualization:** Sarah E. Flanagan, Ana Tangari-Saredo.

**Data curation:** Gabriela Pacheco, Maria G. Bastida, Juan Cáceres, Guillermo Alonso, Mariana Aziz, Martha Suarez, Adriana Flores, Victoria Femenia, María V. Forclaz, Jayne A.L. Houghton, Sabrina Martin, Sarah E. Flanagan, Ana Tangari-Saredo.

**Formal analysis:** Sarah E. Flanagan, Ana Tangari-Saredo.

**Investigation:** Sarah E. Flanagan, Ana Tangari-Saredo.

**Methodology:** Jayne A.L. Houghton, Ana Tangari-Saredo.

**Supervision:** Ana Tangari-Saredo.

**Validation:** Jasmin J. Bennett, Sarah E. Flanagan, Ana Tangari-Saredo.

**Visualization:** Ana Tangari-Saredo.

**Writing – original draft:** Jasmin J. Bennett, Sarah E. Flanagan, Ana Tangari-Saredo.

**Writing – review & editing:** Gabriela Pacheco, Maria G. Bastida, Juan Cáceres, Guillermo Alonso, Mariana Aziz, Martha Suarez, Adriana Flores, Victoria Femenia, María V. Forclaz, Jayne A.L. Houghton, Jasmin J. Bennett, Sabrina Martin, Sarah E. Flanagan, Ana Tangari-Saredo.

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
