## [Decision Letter · Decision Letter 0]

7 May 2025

Thank you for submitting your manuscript to PLOS ONE. After careful consideration, we feel that it has merit but does not fully meet PLOS ONE’s publication criteria as it currently stands. Therefore, we invite you to submit a revised version of the manuscript that addresses the points raised during the review process.

In Table 1 Monosomy X is indicated how was this assessed considering usage of a targeted gene panel?

We look forward to receiving your revised manuscript.

Kind regards,

Klaus Brusgaard

Academic Editor

PLOS ONE

 [This research was funded in whole, or in part, by Wellcome [223187/Z/21/Z]. For the purpose of open access, the author has applied a CC BY public copyright license to any Author accepted Manuscript version arising from this submission.]. 

Additional Editor Comments (if provided):

Reviewers' comments:

Reviewer's Responses to Questions

**Comments to the Author**

1. Is the manuscript technically sound, and do the data support the conclusions?

Reviewer #1: Yes

Reviewer #2: Yes

2. Has the statistical analysis been performed appropriately and rigorously?

Reviewer #1: Yes

Reviewer #2: N/A

3. Have the authors made all data underlying the findings in their manuscript fully available?

Reviewer #1: No

Reviewer #2: Yes

4. Is the manuscript presented in an intelligible fashion and written in standard English?

Reviewer #1: Yes

Reviewer #2: Yes

Reviewer #1: Comments to the Author

The authors summarize genetic analyses of congenital hyperinsulinism (CHI) in Argentina. Given the paucity of reports on CHI-related genetic analyses from South America, this study represents a valuable contribution. The manuscript is generally well-written and presented. However, the reviewers have several questions that should be addressed prior to publication.

Major Comments

P5L129: Did the authors examine the intronic regions of the HK1 gene in all cases that were negative for known pathogenic variants?

P7L174: Did the individuals with late-onset CHI carry any pathogenic variants? Additionally, how was the diagnosis made in the third individual?

P7L178: The reported diagnostic yield appears lower than in other studies. Could the authors clarify the potential reasons for this discrepancy? For instance, is the allele frequency of pathogenic variants lower in this population than in other ethnic groups?

P7L180–181: Reference 36 mentions the 20p11.2 deletion syndrome, yet the current manuscript provides limited molecular genetic details. Similarly, could the authors elaborate further on the preoperative management for the patient with Turner syndrome? Was octreotide trialed? These details would enhance the clarity and clinical relevance of the manuscript.

P10L278: The authors state that access to genetic testing is more limited than 18F-DOPA PET-CT. However, in terms of actual cost, genetic testing is generally considered less expensive. As noted in P5L121, genetic testing was commercially conducted in six patients. Could the authors explain more about access issues in Argentina, such as insurance coverage or national policy implications?

P13–14, Table 1 (Cases 8, 19, 21): The authors state that three patients with diffuse disease underwent surgery, yet the current treatment includes diazoxide in two patients and no treatment in one. Does this imply that diazoxide was effective even postoperatively? A summary of both preoperative and postoperative treatments would improve clarity.

P14, Table 1 (Case 16): Was the use of alpha-glucosidase inhibitors considered for the patient with an INSR variant?

Minor Comments

P8L205: Were the three patients with diffuse disease genetically confirmed cases? If so, providing this information would improve clarity.

P8L218: Please correct the formatting issue—this text portion appears to be unnecessarily italicized.

P13, Table 1: It would be more intuitive to arrange the variants in order of amino acid position (e.g., p.Arg168His before p.Arg598Ter) to facilitate a more straightforward interpretation.

P13, Table 1: Please indicate whether there are any late-onset CHI cases in the table.

Reviewer #2: Tha manuscript is well writen, the objectives and the results are very clear. The presentation of the results is clear and the results were discussied comparing whith other finds at the licterature. The manuscript would be aproved.

**Do you want your identity to be public for this peer review?** For information about this choice, including consent withdrawal, please see our Privacy Policy

Reviewer #1: No

Reviewer #2: **Yes: ** Raphael Del Roio Liberatore Junior

---

## [Author Response · Author response to Decision Letter 1]

6 Jun 2025

RESPONSE TO EDITORIAL COMMENTS

The manuscript is relevant providing insight in the status of CHI in South America. The manuscript is well written. There are some issues that needs to be attended. Please carefully correspond to the reviewer comments and correct accordingly.

Thank you for your support with the review of this manuscript.

In Table 1 Monosomy X is indicated how was this assessed considering usage of a targeted gene panel?

The methods have been updated to clarify the usage of SavvyCNV to perform CNV analysis using on and off target reads (line 162).

RESPONSE TO JOURNAL REQUIREMENTS

The style has been updated throughout to match PLOS ONE’s requirements.

Thank you for stating the following financial disclosure: [This research was funded in whole, or in part, by Wellcome [223187/Z/21/Z]. For the purpose of open access, the author has applied a CC BY public copyright license to any Author accepted Manuscript version arising from this submission.]. Please state what role the funders took in the study. If the funders had no role, please state: "The funders had no role in study design, data collection and analysis, decision to publish, or preparation of the manuscript." If this statement is not correct you must amend it as needed. Please include this amended Role of Funder statement in your cover letter; we will change the online submission form on your behalf.

We have amended the role of funder statement to “This research was funded in whole, or in part, by Wellcome [223187/Z/21/Z]. For the purpose of open access, the author has applied a CC BY public copyright license to any Author accepted Manuscript version arising from this submission. The funders had no role in study design, data collection and analysis, decision to publish, or preparation of the manuscript.”

We note that you have indicated that there are restrictions to data sharing for this study. PLOS only allows data to be available upon request if there are legal or ethical restrictions on sharing data publicly. For more information on unacceptable data access restrictions, please see http://journals.plos.org/plosone/s/data-availability#loc-unacceptable-data-access-restrictions.

We have now added a data availability statement to the manuscript “All non-clinical data analyzed during this study are included in this published article. Clinical and genotype data can be used to identify individuals and are therefore available only through collaboration to experienced teams working on approved studies examining the mechanisms, cause, diagnosis and treatment of diabetes and other beta cell disorders. Requests for collaboration will be considered by a steering committee following an application to the Genetic Beta Cell Research Bank https://www.diabetesgenes.org/current-research/genetic-beta-cell-research-bank/). Contact by email should be directed to rduh.betacellgenomics@nhs.net.”

RESPONSE TO REVIEWER #1

The authors summarize genetic analyses of congenital hyperinsulinism (CHI) in Argentina. Given the paucity of reports on CHI-related genetic analyses from South America, this study represents a valuable contribution. The manuscript is generally well-written and presented. However, the reviewers have several questions that should be addressed prior to publication.

Thank you for taking the time to review our paper and for your constructive comments. We have addressed each of the points you raised, which have helped to strengthen and improve the manuscript.

Major Comments

P5L129: Did the authors examine the intronic regions of the HK1 gene in all cases that were negative for known pathogenic variants?

The non-coding regulatory region in intron 2 of HK1 was screened in all individuals without a disease-causing variant identified. We have now updated the methods to improve clarity and have added the appropriate references:

“For HK1 and SLC16A1, screening was limited to the previously described non-coding regulatory regions where disease-causing variants have been identified.”

P7L174: Did the individuals with late-onset CHI carry any pathogenic variants? Additionally, how was the diagnosis made in the third individual?

Thank you for noting this. One individual with late-onset CHI had a pathogenic GLUD1 variant (case 14 following table reformat). This information has now been highlighted in the text (line 222).

Thank you for highlighting this, we have now rewritten the sentence to clarify that the third patient with late-onset CHI was clinically diagnosed at 8 years old (line 217).

P7L178: The reported diagnostic yield appears lower than in other studies. Could the authors clarify the potential reasons for this discrepancy? For instance, is the allele frequency of pathogenic variants lower in this population than in other ethnic groups?

We believe that the lower diagnostic yield is unlikely to be due to population-specific allele frequencies in known HI genes within the Argentinian population. Rather, we consider it to reflect the composition of the cohort, which includes individuals with transient CHI, later-onset disease, and those with risk factors for perinatal stress hyperinsulinism (PSHI). We have updated the discussion to reflect this interpretation.

“A genetic diagnosis was obtained for 19 of the 49 (39%) patients who underwent testing. This yield is slightly lower than that reported in other large series, likely reflecting the composition of the cohort. Specifically, 7 patients (14%) had transient CHI, 10 individuals had identifiable risk factors for PSHI, including 6 with transient CHI and 4 with persistent HI, and 3 patients had later-onset CHI which is typically associated with a diagnostic yield compared to infancy-onset disease.”

P7L180–181: Reference 36 mentions the 20p11.2 deletion syndrome, yet the current manuscript provides limited molecular genetic details. Similarly, could the authors elaborate further on the preoperative management for the patient with Turner syndrome? Was octreotide trialed? These details would enhance the clarity and clinical relevance of the manuscript.

We have now briefly summarized the clinical details for the patients with 20p11.2 deletion syndrome and Turner syndrome (line 224) and included the reference for the 20p11.2 deletion case in the main text. We have also indicated that the patient with Turner syndrome is included in the description of three of the children with diffuse disease which underwent pancreatectomy due to poor response to diazoxide. We are unable to find any information as to whether octreotide was trialed in this individual.

P10L278: The authors state that access to genetic testing is more limited than 18F-DOPA PET-CT. However, in terms of actual cost, genetic testing is generally considered less expensive. As noted in P5L121, genetic testing was commercially conducted in six patients. Could the authors explain more about access issues in Argentina, such as insurance coverage or national policy implications?

Thank you for highlighting this important point. In comparison, to many high-income countries scanning (imaging) is indeed more accessible than genetic testing in Argentina and in most Latin American countries where imaging techniques such as X-rays, CT scans, MRIs, ultrasounds, and PET scans have a long history in clinical medicine. Their diagnostic value is well-established and directly tied to urgent or actionable decisions, making their use unquestioned by both clinicians and insurers.

In many Latin American countries, health systems are slow to update their coverage catalogues. Since genetic testing is relatively new and rapidly evolving, it’s often not included in official nomenclators (like Argentina’s Nomenclador Nacional), meaning insurers are not obligated to cover them. Imaging procedures, on the other hand, are clearly coded, regulated, and reimbursable, making them easier to approve and fund.

Although imaging can be expensive, it is perceived as cost-effective because it often leads to clear diagnoses or immediate clinical action. Additionally, most clinicians are well-trained in interpreting imaging results, as imaging is an integral part of medical education. In contrast, many physicians in Latin America receive limited training in genetics and may feel uncertain about ordering or interpreting genetic tests, leading to lower demand.

All these aspects are even more pronounced when comparing main urban centres like Buenos Aires to less urban regions. Many patients in this study received care outside the capital, where access to genetic testing is significantly more limited.

We have now included the following paragraph to help qualify our sentence on accessibility to genetic testing.

“In this study, some patients who had not undergone genetic testing underwent scanning in an effort to avoid prolonged medical treatment. In Argentina and most Latin American countries, imaging is generally more accessible than genetic testing, with 18F-DOPA PET-CT introduced in Argentina in 2014. This greater accessibility is largely due to the wider acceptance of imaging for clinical decision-making and its routine coverage by most private health insurance plans. In contrast, genetic testing is usually a non-listed procedure, requiring individual approval from health insurance providers, requests that are often denied. Access is further limited for patients receiving care outside the capital city, where genetic testing resources are even more limited. With the recent availability of charity-funded genetic testing, prioritizing ABCC8 and KCNJ11 screening for future diazoxide-unresponsive cases born in Argentina should be pursued in accordance with current guidelines.”

P13–14, Table 1 (Cases 8, 19, 21): The authors state that three patients with diffuse disease underwent surgery, yet the current treatment includes diazoxide in two patients and no treatment in one. Does this imply that diazoxide was effective even postoperatively? A summary of both preoperative and postoperative treatments would improve clarity.

Thank you for highlight this. We have now updated Table 1 to clarify which individuals were receiving medical treatment post-surgery and which individuals were not receiving medical treatment post-surgery.

P14, Table 1 (Case 16): Was the use of alpha-glucosidase inhibitors considered for the patient with an INSR variant?

We are unable to find any information as to whether alpha-glucosidase inhibitors were considered in this individual.

Minor Comments

P8L205: Were the three patients with diffuse disease genetically confirmed cases? If so, providing this information would improve clarity.

Two of these individuals had disease causing variants and one had a VUS. We have now updated the text to include this information (Line 275).

P8L218: Please correct the formatting issue—this text portion appears to be unnecessarily italicized.

We have updated the formatting of this subheading to comply with the journal style (now line 289).

P13, Table 1: It would be more intuitive to arrange the variants in order of amino acid position (e.g., p.Arg168His before p.Arg598Ter) to facilitate a more straightforward interpretation.

The table has been reformatted in order of amino acid position and the case numbers have been updated.

P13, Table 1: Please indicate whether there are any late-onset CHI cases in the table.

A genetic variant was identified in one of the three cases with late onset CHI. They are case 14 in the table and we have now updated the text to indicate this (line 222).

REVIEWER 2:

Tha manuscript is well writen, the objectives and the results are very clear. The presentation of the results is clear and the results were discussied comparing whith other finds at the licterature. The manuscript would be aproved.

We thank the reviewer for the time their time and are pleased that they found our paper suitable for publication.

---

## [Decision Letter · Decision Letter 1]

8 Jul 2025

Characterization of congenital hyperinsulinism in Argentina: clinical features, genetic findings, and treatment outcomes

PONE-D-25-10972R1

Dear Dr. Sarah Flanagan,

We’re pleased to inform you that your manuscript has been judged scientifically suitable for publication and will be formally accepted for publication once it meets all outstanding technical requirements.

Kind regards,

Klaus Brusgaard

Academic Editor

PLOS ONE

Additional Editor Comments (optional):

Reviewers' comments:

Reviewer's Responses to Questions

**Comments to the Author**

Reviewer #1: All comments have been addressed

Reviewer #2: All comments have been addressed

2. Is the manuscript technically sound, and do the data support the conclusions?

Reviewer #1: Yes

Reviewer #2: Yes

3. Has the statistical analysis been performed appropriately and rigorously?

Reviewer #1: Yes

Reviewer #2: Yes

4. Have the authors made all data underlying the findings in their manuscript fully available?

Reviewer #1: No

Reviewer #2: Yes

5. Is the manuscript presented in an intelligible fashion and written in standard English?

Reviewer #1: Yes

Reviewer #2: Yes

Reviewer #1: (No Response)

Reviewer #2: (No Response)

**Do you want your identity to be public for this peer review?** For information about this choice, including consent withdrawal, please see our Privacy Policy

Reviewer #1: **Yes: ** Shinji Higuchi

Reviewer #2: **Yes: ** Raphael Del Roio Liberatore Junior

---

## [Editor Report · Acceptance letter]

PONE-D-25-10972R1

PLOS ONE

Dear Dr. Flanagan,

I'm pleased to inform you that your manuscript has been deemed suitable for publication in PLOS ONE. Congratulations! Your manuscript is now being handed over to our production team.

Kind regards,

on behalf of

Dr. Klaus Brusgaard

Academic Editor

PLOS ONE